# The Antecedents of Customer Satisfaction in the Portuguese Telecommunications Sector



José Torrão [1] and Sandrina Teixeira [2,*]

1   FEP, School of Economics and Management, University of Porto, 4200-464 Porto, Portugal
2   CEOS.PP, ISCAP, Polytechnic of Porto, 4465-004 Porto, Portugal
*   Correspondence: sandrina@iscap.ipp.pt

**Abstract:** This study's primary goal is to examine the elements that affect customer loyalty and satisfaction with Portuguese telecommunications. Indeed, customer loyalty and satisfaction are crucial factors in guaranteeing the success and expansion of the services sector. Furthermore, it aims to include customers' privacy perceptions in a thorough model. A structured questionnaire was adapted from previous studies in the field, collecting a total of 357 valid responses. The suggested hypotheses were tested using multiple statistical techniques to assess the reliability and validity of the gathered data, culminating with path analysis through Structural Equation Modelling. The research results demonstrate that consumer loyalty is highly impacted by satisfaction. On the other hand, service quality significantly influences customer satisfaction, whereas trust and perceived value have a positive yet insignificant impact on this construct. Additionally, perceptions of privacy risk were found to affect customer trust positively and significantly. Considering that the data used for this analysis were collected exclusively in the Portuguese market, inferring the same findings in different countries should be made prudently. As this study only comprised of one of the perceived value dimensions, the results associated with this construct should also have that in mind.

**Keywords:** customer loyalty; customer satisfaction; service quality; trust; perceived value; privacy risk; telecommunications

## 1. Introduction

The business world consistently deals with numerous challenges as competitors arise, and new technologies are developed in different industries. Such a scenario leads companies to embrace innovative strategies that bring added value to their customers, ultimately developing a unique competitive advantage and strengthening ties with the stakeholders. This approach is valid, not only from a product point of view but also from a service frame, as some authors point out that the latter is being increasingly prioritised over the former in the marketing literature [1]. In fact, over the past decades, multiple reasons have contributed to steady growth in the overall market competition [2], offering customers seemingly limitless options for goods and services.

The telecommunications sector is characterised by fierce competition among its players as market penetration of some of its services frequently exceeds 100 per cent of the population in various countries [3]. Therefore, boosting current customers' value is a major issue for service providers. This context highlights the importance of putting customers' needs first and developing long-lasting relationships with them [4]. Analysing customer loyalty and its antecedents in these circumstances emerge as a critical success factor for companies. As a result, service providers are increasingly focusing on delivering top-quality services to their customers, promoting satisfaction, and earning their trust [5].

In Portugal, telecommunications is a mature market with four significant players: Altice Portugal, NOS, Vodafone and NOWO. The sector is gradually leaning toward bundling services as ANACOM [6] estimates that, at the end of the first quarter of 2022, the residential penetration of these services had reached 89.6 out of 100 households, representing a

3.67% average annual growth rate in the past five years. The bundling services of the four mentioned operators represented revenue of approximately 4.46 billion euros in 2021 [6]. Even if the importance of telecommunications in the country is unquestioned, national and international institutions' reports usually indicate a high concentration [6,7]. Considering data from ANACOM [6], a small number of operators combined with a substantial market share from three led to an HHI value consistently above 2500, thus indicating high concentration according to the U.S. Department of Justice and the Federal Trade Commission [8]. This scenario implies a rather low competition and hence might be one of the reasons behind the relatively high prices this sector has in the country [7].

The Portuguese telecommunications market is, indeed, a unique one. During the first semester of 2022, the "Metadata Law" was in the media's light following a government's proposal to change some of its terms. Such a setting brought data privacy issues to the public debate, and multiple concerns arose as this bill contained clear orientations for service providers to preserve data regarding location, time and equipment used in communications throughout one year. Telecom operators' devotion to ensuring the privacy of their clients' information is critical at this point. Such circumstances call for an analysis of customers' perceptions of the effort service providers put into this subject, as it might translate into reinforced or decreased trust in a given company, ultimately impacting the duration of the customer-firm relationship [9].

In the past, numerous studies analysed customer loyalty and its antecedents in different market fields. The same subject was also extensively studied within the telecommunications market, and multiple constructs were put into various perspectives. Still, there seems to be a gap in the literature when considering the inclusion of privacy risk perceptions in such models. This study aims to fill this gap through a comprehensive conceptual model examining customer loyalty, satisfaction, trust, service quality, perceived value, and privacy risk, hence delivering strong insights both to research and managers.

## 2. Literature Review

### 2.1. Customer Satisfaction and Customer Loyalty

Kim et al. [10] referred to customer satisfaction through a perspective that focused on consumers' thoughts regarding a post-purchase scenario compared to their initial expectations. Satisfaction is a comprehensive emotion influenced by service quality, pricing, and contextual or personal circumstances [11]. Therefore, it is seen as a vital and decisive aspect of repurchasing a product or acquiring a service, particularly an intangible one [12]. Such a paradigm requires companies to consider customer satisfaction as a significant matter while developing strategies to promote it and create value for their business. This condition is reinforced by Fornell [13], pointing out that customer satisfaction positively influences loyalty and leverages customer retention and acquisition by lowering price sensitivity and reducing costs. Loyal customers are associated with an increased likelihood of future purchases, as well as being more inclined to raise their spending within the company and recommend the brand via positive word-of-mouth [14–16].

Previous research in the marketing field has been keen on establishing customer satisfaction as a critical precursor of customer loyalty [13,17]. This relationship is crucial to corporate management because it represents effective marketing programmes [18] that ultimately influence a company's financial success [19]. Concerning the telecommunications market, the former investigation has steadily recognised that customer satisfaction does promote post-purchase perceptions and actions, ultimately leading to increased customer loyalty [20–23]. Hence, the first hypothesis of this study is suggested as follows:

**Hypothesis 1.** *Customer Satisfaction has a major influence on Customer Loyalty.*

## 2.2. Trust and Customer Satisfaction

Trust is regarded as one of the most meaningful antecedents of secure and collective partnerships in business [24]. The former investigation has proposed that trust consists of the assumption that an individual in a business relationship would behave in their partner's best interest [25–29] and the sense of integrity recognized between the individuals or groups involved [30]. Thus, this construct is determined by the customer's expectations of whether the service provider is reliable and follows through on its promises [31].

There is evidence to suggest that trust can lead to customer satisfaction [32–34]. Trust is defined as the belief in the reliability, truth, ability, or strength of someone or something [35]. When customers trust a company or product, they are more likely to be satisfied with their experience because they have confidence in the product or service and believe it will meet their needs [36]. Kassim and Abdullah [37] have established a link between customers who trust their service provider and their satisfaction. Such a relationship exploits this construct's relevance as a customer lacking confidence in its service provider will almost certainly be unsatisfied. Concerning the Theory of Reasoned Action, it is also acknowledged that trust promotes satisfaction, which ultimately enhances loyalty [5]. Other authors, such as Rasheed and Abadi [38] or Park et al. [39] were also prone to set a strong relationship where trust leads to customer satisfaction. In fact, the latter emphasized this link in an analysis where it was verified and accepted that trust in a service provider is a crucial antecedent of customer satisfaction in mobile commerce. Hence, the second hypothesis of this study is suggested as follows:

**Hypothesis 2.** *Trust has a positive effect on Customer Satisfaction.*

## 2.3. Service Quality, Customer Satisfaction and Trust

Quality can be defined as customers' notion of the value of services in a post-purchase scenario, providing insights to the firm on whether their services are valuable [40]. Other authors define service quality as an attribute that concerns reliability, dependability, trustworthiness, and responsiveness [41]. Regarding a possible link between service quality and customer satisfaction, some authors suggest that a customer's assessment of the former represents a customer's level of satisfaction with their post-purchase perception of the service [42]. Khan and Fasih [43] also agree that service quality significantly influences a customer's perception of a given service. An increased level of service quality promotes customer satisfaction and impacts consumers' purchase behaviour [44]. This construct is also critical for success over time and gaining a competitive advantage [45], therefore, being a key indicator of customer satisfaction concerning service providers' efficiency [20,46].

Research has consistently shown a positive correlation between service quality and customer satisfaction [47–49]. Previous studies in the telecommunications field have also found that service quality is an essential predictor of customer satisfaction [50,51]. Overall, service quality plays a crucial role in determining customer satisfaction, which is why businesses must focus on delivering high-quality service to satisfy their customers. Hence, the third hypothesis of this study is suggested as follows:

**Hypothesis 3.** *Service Quality has a positive effect on Customer Satisfaction.*

Considerable research in the marketing field has attempted to establish a link between service quality and trust [52]. There are several ways in which service quality can lead to trust. When customers receive services that meet or exceed their expectations, they are more likely to develop trust in the service provider [34]. Uzir et al. [34] also add that this is because customers feel that their needs are being met and that the service provider is reliable and competent.

Several empirical studies have analysed the direct relationship between quality and trust [48,53]. Gounaris and Venetis [54] were inclusively able to establish that the degree to which a customer trusts their service provider is influenced by service quality. Indeed,

service providers promote specific offerings to assure their clients' trust and to develop a relationship of confidence with them [55]. For instance, concerning the 5G launch, service providers worldwide promoted free trial packages where customers could assess the quality of the fifth-generation mobile network for a limited time. Such strategies promote customer trust in a company's dependability [56] and are likely to increase confidence in the service provider [57]. In a nutshell, considering that trust relates to consumers' views of a company's reputation, credibility, and ability to meet expectations [58], it is tightly linked to service quality, making customers more inclined to trust a service provider that improves overall service quality [54]. Hence, the fourth hypothesis of this study is suggested as follows:

**Hypothesis 4.** *Service Quality has a positive effect on Trust.*

*2.4. Perceived Value, Customer Satisfaction and Trust*

The concept of perceived value has been studied in many different circumstances [59–61], and some authors inclusively state that its study has dominated the services literature [62]. Despite the introduction of numerous conceptual models of value [60,63], perceived customer value has frequently been defined as the trade-off between what is received and provided by consumers when acquiring a service [64–66]. In the same line of thought, Colorado and Mesias [1] suggest this construct represents the exercise customers make when setting different purchasing options side by side as well as the judgement of the utility and cost of each option. It is also important to mention that multiple authors advocate that value measurement depends on different factors, such as service type, situational conditions, previous experiences, and client attributes [67,68]. As a result, the definition of value potentially differs among customers [69].

As Kim and Kang [70] also posit that human behaviour is strongly linked to a comprehensive comparison of what is given and received, they conclude that perceived value is composed of four dimensions: functional, emotional, monetary, and social value. Zeithaml likewise proposes a multidimensional view of this construct, stating, "(1) value is low price, (2) value is whatever I want in a product, (3) value is the quality I get for the price I pay, and (4) value is what I get for what I give." [71] (p. 13). Other academics also differentiate functional and symbolic value concepts [71,72]. According to Lai et al. [64], functional value entails broad assessments of quality and value for money. On the other hand, Zeithaml [71] adds that it regards how customers evaluate the quality of the goods and services offered, their purchase price and the time sacrificed for the purchase. Contrarily, symbolic value denotes impressions of past experiences regarding community, feelings, aesthetics, and reputation [72]. Customers are not indifferent to societal opinions, which consist of an external influence on the symbolic value that is also comprised of an internal sense of desire and delight [73]. As far as this study is concerned, value is analysed as functional value with a particular focus on price and value for money since there is already a specific construct to analyse the perceptions of service quality.

Regarding a possible relationship between perceived value and customer satisfaction, McDougall and Levesque [74] state the importance of reaching the bottom of this potential link. In previous research on this topic, empirical studies of traditional retailers suggest that perceived value is likely to affect customer satisfaction positively [60,62,75]. Identical results were also produced in e-commerce [76,77] and multiple telecommunications markets worldwide [78–81]. Hence, the fifth hypothesis of this study is suggested as follows:

**Hypothesis 5.** *Perceived Value has a positive effect on Customer Satisfaction.*

Trust usually results from a brand or company's ability to fulfil its promises [82]. Consequently, building and maintaining relationships in various trade scenarios depends on trust [83]. Due to the intangible character of services, which bears a sense of unpre-

dictability for customers through purchase and consumption, it is especially pointed out that a service relationship with a client depends on trust [84,85].

Concerning the link between perceived value and trust, multiple authors posit that these two constructs have a positive connection [86,87].

Indeed, some empirical studies propose that trust assessments impact perceived value through customers' continuous interactions with service providers [31]. Nevertheless, this relationship is mainly regarded in line with Harris and Goode's view, which state that "trust is a key and central factor during exchange, after accounting for previously established antecedents, namely; perceived value" [86] (p. 150). Other studies have reached the same conclusion on this subject [88,89], inclusively in the telecommunications field [90]. Hence, the sixth hypothesis of this study is suggested as follows:

**Hypothesis 6.** *Perceived Value has a positive effect on Trust.*

*2.5. Privacy Risk and Trust*

The concept of privacy risk is an increasingly debated topic among researchers. Featherman et al. [91] establish privacy risk as the outcome of research on information privacy [92,93] and perceived risk [94–96] and define it as customers' perceptions of potential losses. Additionally, the authors note that this construct is based on an individual's evaluation of the probability of information misuse and data loss, which may eventually harm clients' privacy.

Information privacy has also progressively emerged as a significant concern for customers and is characterised as "the claim of individuals, groups, or institutions to determine for themselves when, how, and to what extent information about them is communicated to others" [93] (p. 7). According to research, consumer privacy issues are pervasive, rising, and may worsen in the future [97]. Such situations are undoubtedly crucial in the digital era [98,99].

On the other hand, Perceived risk concerns customers' uncertainty regarding the outcome of their decisions [100]. Cox and Rich [101] assert that negative outcomes and uncertainty are decisive components of perceived risk. A customer may experience risk when purchasing or dealing with uncertainty and unfavourable outcomes [94,102]. As a result, if the outcomes were unfavourable, clients would sacrifice money, time, and other potential damage [103]. According to Jacoby et al. [104], consumers may acknowledge different risks, including the operational, physical, financial, social, psychological, and general perceptions of risk. Zhang et al. [102] developed and validated more aspects of perceived risk, including social, economic, privacy, time, quality, health, delivery, and after-sale risks. This study focuses on studying perceived risk in terms of privacy.

Considering the two previously explained constructs, research has shown that the perception of privacy risk can influence trust [105–108]. Trust is a critical factor in determining how people interact with each other and with institutions [109], and the perception of privacy risk can affect trust in several ways. For example, if people feel that their privacy is being violated or that their personal information is at risk of being misused, they may be less trusting of the organisation collecting or handling that information [110]. Furthermore, if people feel that their privacy is being respected and that their personal information is being handled responsibly, they may trust the organisation or individual in question more [111]. In general, the perception of privacy risk is an essential factor that can influence trust and the willingness of people to share personal information with others [112]. Hence, the final hypothesis of this study is suggested as follows:

**Hypothesis 7.** *Privacy Risk has a positive effect on Trust.*

## 3. Materials and Methods

The chosen methodology for developing the investigation emphasises the research objectives and defines the model testing the proposed hypotheses analysed in the previous chapter, as displayed in Figure 1.

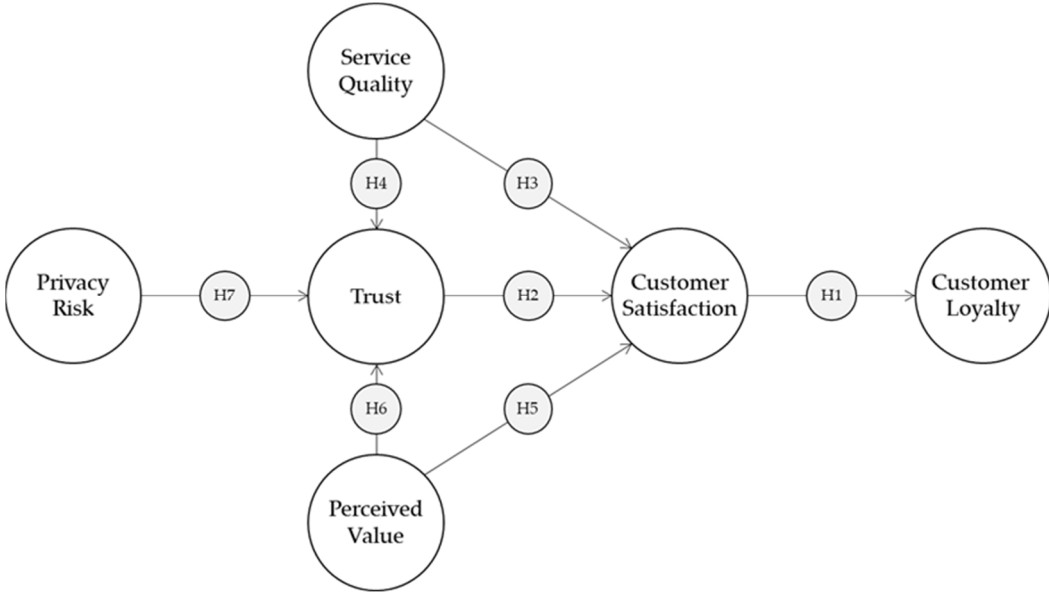

**Figure 1.** Proposed conceptual model.

### 3.1. Survey and Measurements

This study required the elaboration of a questionnaire contained in a survey that was divided into two major sections: the first consisted of items measuring the research variables, and the second one regarded customer profile characteristics. To ensure well-grounded results, the measurement items in the questionnaire followed previously validated studies from the literature.

Customer Loyalty (CL) followed Morgan and Govender's study [113], containing three items (CL1: I am loyal to my service provider; CL2: I will not switch my service provider; CL3: If I was starting again, I would choose my current service provider again as my main service provider).

Customer Satisfaction (CS) was likewise adapted from Morgan and Govender [113] and comprised of three items (CS1: Considering everything, I am satisfied with my service provider; CS2: My service provider always meets my expectations; CS3: I feel that my service provider gives me exactly what I need).

Trust (TR) was adjusted from Aydin and Ozer [114], consisting of four items (TR1: I trust this company; TR2: I feel that I can rely on this company to serve well; TR3: I trust the billing system; TR4: I believe that I can trust this company will not try to cheat me).

Service Quality (SQ) was adapted from Morgan and Govender [113] and included three items (SQ1: My service provider has an excellent service quality; SQ2: The network coverage/reception is good; SQ3: The internet speeds are fast).

Perceived Value (PV) was measured in accordance with research from Morgan and Govender [113] and comprised of three items (PV1: I get value for money with my service provider; PV2: The tariffs and fees at my service provider are fair; PV3: My service provider has good prices and promotions compared to competitors).

Finally, Privacy Risk (PR) was adapted from Taylor, Ferguson and Ellen [115], and consisted of four items (PR1: Keeping my personal information and activities confidential is a high priority for my service provider; PR2: My service provider regards information about my personal life as a strictly private matter; PR3: Guarding my personal information

is one of the highest priorities of my service provider; PR4: Overall, my service provider has a strong need to protect my personal information).

Since this study targets the Portuguese market, the survey was available exclusively in Portuguese. This scenario demanded a translation through the retro-translation method in which the items were first translated into Portuguese and were later translated back into English by a different individual, ultimately comparing the obtained and original items. It was also guaranteed a minimum of three items for each of the analysed constructs, which, according to Hair et al. [116], provides estimates with a higher level of confidence. Still, it was considered crucial to shorten the total number of measures as Schmitt and Stults [117] state it is an effective way to reduce potential exhaustion or distraction from respondents, ultimately leading to somewhat biased results. Finally, all items were measured using a seven-point Likert scale where 1 indicates strong disagreement and 7 indicates strong agreement.

### 3.2. Data Collection and Sample Description

As was previously stated, the primary research tool consisted of a structured questionnaire. The responses to this questionnaire were collected in a survey developed on Google Forms web-based software after a pre-test was carried out with a small sample of respondents. This process allowed the optimization and assessment of the understanding of each item and the questionnaire. The final version of the survey was subsequently shared through e-mail and social media, collecting a total of 357 responses gathered between 24 May and 6 June 2022—all the responses were valid and consisted of a convenience sample. Each questionnaire had an average answering time of approximately 3 min, comprising of both sections of the survey. As was previously stated, after respondents provided the answers to the measurement items, the second section of the survey included questions regarding their demographic characteristics. This section delivered information on the variety of the sample and ensured the collected responses were diverse despite using a convenience sampling method.

An altogether characterization of the sample is summarised in Table 1.

### 3.3. Reliability and Validity

This study involved performing several statistical tests to assess the data's reliability and validity before the hypotheses testing could be performed through Structural Equation Modelling (SEM). The process included an Exploratory Factor Analysis (EFA), followed by a Confirmatory Factor Analysis, among other statistical techniques that promoted the verification and optimisation of the measurement model. The software tools used to conduct these analyses were IBM SPSS 27 and IBM AMOS 28.

Prior to the conduction of an EFA, the Kaiser-Meyer-Olkin measure for sampling adequacy (KMO) and Bartlett's sphericity test were performed to assess the data's suitability for factor analysis. On the one hand, an overall KMO of 0.945 may be evaluated as marvellous [118] and significantly above the recommended minimum of 0.600 [119]. This number suggests a high proportion of variance among the variables derived from the systematic or common variance and, thus, an appropriate sample for factor analysis. On the other hand, Bartlett's sphericity test indicates a significance level of 0.000, revealing that the correlation matrix differs from the identity matrix. The commonalities were also all above 0.600. This scenario reinforces the adequacy of factor analysis. Concerning EFA, Principal Axis Factoring was the chosen extraction method combined with Promax rotation. The factor loadings of the items in the study ranged from 0.532 to 0.952, above the cut-off value of 0.500 [116], as seen in Table 2. Regarding the internal consistency of the variables, all factors revealed fairly high Cronbach's alpha [120], as demonstrated in Table 2. These values suggest the high reliability of the items measuring each of the dimensions in the study. In what comes to item-total correlations, its values ranged from 0.554 to 0.858, also above the usually recommended value of 0.400.

**Table 1.** Demographic Characteristics.

| Variables | Categories | Frequency | Percentage |
|---|---|---|---|
| Gender | Female | 211 | 59.1% |
| | Male | 146 | 40.9% |
| Age range | 18 to 24 | 76 | 21.3% |
| | 25 to 34 | 35 | 9.8% |
| | 35 to 44 | 39 | 10.9% |
| | 45 to 54 | 83 | 23.2% |
| | 55 to 64 | 98 | 27.5% |
| | 65 or more | 26 | 7.3% |
| Academic degree | Elementary School | 0 | 0.0% |
| | Middle School | 4 | 1.1% |
| | High School | 74 | 20.7% |
| | Bachelor Degree | 143 | 40.1% |
| | Post-graduate | 32 | 9.0% |
| | Master's Degree | 83 | 23.2% |
| | PhD | 17 | 4.8% |
| | Other | 4 | 1.1% |
| Fibre coverage in residence area | My current service provider has fibre coverage in my area. | 321 | 89.9% |
| | Other service providers, but not my current one, have fibre coverage in my area. | 30 | 8.4% |
| | No service provider has fibre coverage in my area. | 6 | 1.7% |
| Professional situation | Student | 61 | 17.1% |
| | Employed | 243 | 68.1% |
| | Unemployed | 4 | 1.1% |
| | Retired | 24 | 6.7% |
| | Other | 25 | 7.0% |
| Household size | 1 person | 32 | 9.0% |
| | 2 persons | 81 | 22.7% |
| | 3 persons | 103 | 28.9% |
| | 4 persons | 107 | 30.0% |
| | 5 or more persons | 34 | 9.5% |
| Household net monthly income | Up to 750 € | 15 | 4.2% |
| | From 750 € to 1500 € | 53 | 14.8% |
| | From 1500 € to 2250 € | 77 | 21.6% |
| | From 2250 € to 3000 € | 73 | 20.4% |
| | More than 3000 € | 139 | 38.9% |

After conducting an EFA, a Confirmatory Factor Analysis (CFA) was performed to assess the validity of the latent variables [121]. The execution of the EFA and CFA resulted in eliminating three of the 23 items included in the questionnaire. The CFA allowed measuring the level to which the collected data suited the measurement model. It also assessed the validity of the remaining 20 items in the measurement model before analysing the relationships of the variables in the structural model. In other words, CFA aims to evaluate the construct validity of a given measurement theory [116]. Construct validity is usually assessed by analysing convergent and discriminant validity for each latent variable [122]. The first can be defined as the property of items related to a particular construct. These items typically converge or reveal a significant fraction of variance in common [116]. Among the indicators that are usually pointed out as relevant to evaluate convergent validity are factor loadings. In this study, all items were statistically significant as they loaded above 0.500 [116]. Furthermore, Table 2 presents other insights on convergent validity with Composite Reliability (CR) and Average Variance Extracted (AVE). The former illustrates an aggregate view of the reliability of each construct and should have a value of

at least 0.600 [123], although more recent research suggests a minimum value of 0.700 [116]. The latter represents the share of variance seized by the construct compared to variance related to measurement error and should have a value of no less than 0.500 [122]. As Table 2 suggests, all the analysed constructs depict fair values, most of which are significantly above the minimum recommended.

**Table 2.** Reliability and validity.

| Construct | Item | Cronbach's Alpha | CR | AVE | EFA Loading | CFA Loading |
|---|---|---|---|---|---|---|
| Customer Loyalty | CL1 | | | | 0.801 | 0.537 |
| | CL2 | 0.776 | 0.789 | 0.564 | 0.876 | 0.675 |
| | CL3 | | | | 0.532 | 0.802 |
| Customer Satisfaction | CS1 | | | | 0.859 | 0.859 |
| | CS2 | 0.902 | 0.937 | 0.833 | 0.886 | 0.885 |
| | CS3 | | | | 0.870 | 0.871 |
| Trust | TR1 | | | | 0.848 | 0.870 |
| | TR2 | 0.905 | 0.928 | 0.763 | 0.874 | 0.909 |
| | TR3 | | | | 0.786 | 0.750 |
| | TR4 | | | | 0.849 | 0.775 |
| Service Quality | SQ1 | | | | 0.717 | 0.905 |
| | SQ2 | 0.891 | 0.911 | 0.773 | 0.906 | 0.763 |
| | SQ3 | | | | 0.914 | 0.747 |
| Perceived Value | PV1 | | | | 0.856 | 0.887 |
| | PV2 | 0.898 | 0.902 | 0.754 | 0.918 | 0.876 |
| | PV3 | | | | 0.823 | 0.830 |
| Privacy Risk | PR1 | | | | 0.859 | 0.866 |
| | PR2 | 0.954 | 0.898 | 0.688 | 0.923 | 0.920 |
| | PR3 | | | | 0.952 | 0.946 |
| | PR4 | | | | 0.931 | 0.946 |

On the other hand, discriminant validity tests whether concepts are unrelated, even if they share similarities [116]. As is demonstrated in Table 3, the values for squared correlations between all constructs are below values for AVE, granting the existence of discriminant validity in this study.

**Table 3.** Discriminant Validity.

| Factor | CL | CS | TR | SQ | PV | PR |
|---|---|---|---|---|---|---|
| CL | 0.564 | 0.301 | 0.343 | 0.295 | 0.184 | 0.495 |
| CS | 0.548 | 0.833 | 0.424 | 0.485 | 0.549 | 0.543 |
| TR | 0.586 | 0.651 | 0.763 | 0.248 | 0.156 | 0.397 |
| SQ | 0.543 | 0.696 | 0.498 | 0.773 | 0.312 | 0.527 |
| PV | 0.429 | 0.741 | 0.394 | 0.558 | 0.754 | 0.388 |
| PR | 0.703 | 0.737 | 0.630 | 0.726 | 0.623 | 0.688 |

Note: Below the diagonal—correlations between variables; Above the diagonal—squared correlations between variables; Diagonal—AVE.

## 4. Results

Both the measurement and structural models' goodness-of-fit should be assessed concerning multiple measures, including indices of absolute fit, incremental fit, goodness-of-fit, and badness-of-fit [116]. As Table 4 suggests, all the values evaluating goodness-of-fit within the measurement and structural models indicate acceptable model fit for all indices following the recommended values from Hair et al. [116]. The listed recommended numbers considered the authors' revision from previous studies and have in mind the sample size and number of observed variables in this study—357 responses and 20 observed variables.

**Table 4.** Measurement and Structural Models.

| Fit Indices | Recommended Value | Measurement Model | Structural Model |
|---|---|---|---|
| $\chi2/df$ | <3.000 | 2.568 | 2.520 |
| RMSEA | <0.070 | 0.066 | 0.065 |
| GFI | >0.900 | 0.904 | 0.903 |
| CFI | >0.940 | 0.966 | 0.966 |
| TLI | >0.940 | 0.956 | 0.958 |

Considering the structural model revealed a satisfactory fit, the analysis then proceeded to estimate the path coefficients between variables, confirming or rejecting this investigation's suggested hypotheses. Table 5 portrays the results for the seven hypotheses in this study, including standardised estimates, standard error, critical ratio, significance level, and result of approval. Except for H4 and H6, all hypotheses were significant ($p < 0.001$) and consequently accepted. The following subsections analyse the hypotheses' results in detail.

**Table 5.** Results of the hypotheses test.

| Hypothesis (Path) | β | S.E. | C.R. | *p*-Value | Result |
|---|---|---|---|---|---|
| H1 (CL ← CS) | 0.961 | 0.073 | 13.210 | *** | Confirmed |
| H2 (CS ← TR) | 0.101 | 0.077 | 1.316 | 0.188 | Rejected |
| H3 (CS ← SQ) | 0.701 | 0.082 | 8.540 | *** | Confirmed |
| H4 (TR ← SQ) | 0.466 | 0.059 | 7.915 | *** | Confirmed |
| H5 (CS ← PV) | 0.055 | 0.052 | 1.055 | 0.291 | Rejected |
| H6 (TR ← PV) | 0.275 | 0.054 | 5.081 | *** | Confirmed |
| H7 (TR ← PR) | 0.335 | 0.039 | 8.537 | *** | Confirmed |

Note: *** ($p = 0.000$).

The results from Table 5 indicate the acceptance of H1 with solid support (H1: β = 0.961; $p = 0.000$), establishing customer satisfaction as the primary driver of customer loyalty in this study. Such a conclusion is corroborated by previous research in the field. Indeed, Kim et al. [20] concluded that highly satisfied customers tend to stay with their current service providers and keep their subscriptions. Multiple studies firmly confirm the relationship between customer satisfaction and customer loyalty in the literature, inclusively in the telecommunications sector [1,5,21,50,90,113,124,125].

Analysing the outcomes for H2, the numbers suggest its rejection (H2: β = 0.077; $p = 0.188$). In fact, there is a positive yet frail and insignificant relationship between trust and customer satisfaction, thus not supporting this hypothesis. Contrary to what the reviewed literature suggests, where the link between these two dimensions was frequently supported [5,37,38,126,127], it is essential to take into consideration the particularities of the Portuguese telecommunications market. As was previously mentioned, this sector is characterised by a high concentration, making it somewhat less competitive as the three major players tend to adopt similar behaviours. This scenario might explain the insignificance of some constructs in promoting the satisfaction of Portuguese customers. Indeed, as the main service providers in the country have identical ways of conduct, trust does not reveal to be a key determinant of customers' satisfaction, having a rather neutral impact on it.

The empirical data also aligned with the hypothesised link between service quality and customer satisfaction, thus confirming H3 (H3: β = 0.701; $p = 0.000$). Even if few researchers found this relationship inconclusive [113], this scenario seems to be an exception to most of the previously made analyses. Therefore, with support from the literature [1,5,50,55,128], this study concludes that customers' assessment of a service's quality reflects their satisfaction with that service.

Similarly, values in Table 5 point out the confirmation of H4, establishing a connection in which service quality promotes trust among customers (H4: $\beta$ = 0.466; *p* = 0.000). Once again, other authors also hypothesised and confirmed this same tie in the past [1,55,114], suggesting this is a crucial relationship in the telecommunications market around the world.

This study's findings indicate that the link between perceived value and customer satisfaction is not supported (H5: $\beta$ = 0.055; *p* = 0.291). Even if there is a positive relationship between these two constructs, the connection is insignificant, conversely to what is often verified in the marketing literature [1]. Therefore, a justification for this result might lean predominantly on two factors. Firstly, as was noted before, this study regards perceived value from a functional point of view, emphasising price and value for money. Considering this assumption, the rejection of H5 aligns with findings from Kim et al. [20], where the effect of pricing structure on customer satisfaction was not statistically verified, concluding that the former has little to no impact on the latter. A second reason for this outcome might be related to the concentration of the Portuguese telecommunications market. According to OECD [7], broadband prices are reasonably high, and service providers have no incentive to change them, as competition is low due to high concentration. According to the referred assumption, this situation might reinforce the perceived value's negligibility on satisfaction.

Oppositely, the gathered data implies the acceptance of H6 (H6: $\beta$ = 0.275; *p* = 0.000), hence confirming a positive link between perceived value and trust. This finding is in line with the literature, as this relationship has consistently been confirmed over the years. In fact, studies in multiple fields, including the telecommunications sector, have reached similar conclusions [1,90].

Table 5 suggests that customers' perceptions of privacy risk positively affect trust, confirming H7 (H7: $\beta$ = 0.335; *p* = 0.000). Confirmation of this relationship is fundamental for this analysis, as there was a relatively meagre study of this particular link within the telecommunications field. Still, the findings are in line with conclusions from Libaque-Saenz et al. [129], in which multiple dimensions of privacy and information risks are related to trust and evaluated in a thorough model.

## 5. Conclusions

The Portuguese telecommunications market is well-developed due to the continuous investment made by the major service providers in the past decades. Despite institutions' statements referring to its relatively low competition, the country usually ranks among the best in Europe in terms of broadband capacity and high-speed internet, which covers most of the country. As was previously mentioned, the characteristics of this market make it unique.

This study's primary goal was to combine the most critical determinants of customer loyalty and satisfaction in the telecommunications sector, considering the vast literature in the field to select the necessary constructs. Thus, the impact of service quality, trust and perceived value on customer satisfaction was hypothesised and estimated. Furthermore, it aimed to develop a model where customers' perceptions of privacy risk were included since the subject is an increasing point of focus worldwide and its relevance in Portugal is currently more significant following the public debate on the "Metadata Law". The links between variables were assessed through structural equations after the reliability and validity of the data were confirmed. Despite having negligibly different objectives, few investigations have been developed in Portugal with a similar approach to loyalty in telecommunications [130,131]. This study's considerable sample size and the multiple tests performed on the questionnaire, its measurements and constructs contribute to high confidence in the trustworthiness of this study's findings. Hence, this investigation represents a notable addition to the literature as it evaluated the suggested hypotheses in the sector. Not only does it confirm some major accepted views on the subject, but it also establishes the differences the field has in the country compared to other nations. Notably, the insignificance of trust and perceived value on customer satisfaction were majorly explained by market particularities. Still, the prominence of service quality in driving satisfaction

in this field was also demonstrated by Kim et al. [20] through a descriptive statistical analysis of empirical data. Furthermore, this study provides unique insights into how the construct of privacy risk relates to trust, as this relationship in the telecommunications market was scarcely analysed in the past. In fact, customers' perceptions of privacy risk were demonstrated to drive trust among Portuguese clients with solid support.

This study's findings also provide critical information for managers in the Portuguese telecommunications field. As the market is already mature, service providers should adapt their strategies to promote longer customer-firm relationships, increasing loyalty. The analysis was keen to conclude that satisfaction is undoubtedly the primary driver of customer loyalty. Furthermore, the results demonstrated that satisfaction is more likely to be explained by service quality, suggesting that the head of organisations should focus on building methods that promote it. Such strategies can deliver a sustainable competitive advantage for mobile operators if implemented successfully. For companies aiming to foster trust among their clients, this study's main conclusions also suggest the importance of emphasising the dedication to preserving customers' data safely. Indeed, their perceptions on this matter influence trust in a service provider. Therefore, telecommunications companies should protect users' data through transparent policies on how they store it and under which circumstances they are allowed to use it.

Despite its contributions, this study is not without limitations that might require some consideration in the analysis of its findings. Certainly, there are no absolute truths. The first limitation is related to the sample used in this study. Despite trying to maximise its randomness and diversity, the results only refer to this sample and comprise the Portuguese market. Therefore, its generalisation to other countries should be made carefully. In addition, it considers solely one of the various dimensions of perceived value. Future research can explore this construct within its multiple extents. Thirdly, this study did not examine a potential mediating role of satisfaction in an indirect relationship between service quality, trust, and perceived value on customer loyalty. Similarly, this subject is suggested to be studied in further research. Finally, as there was a scarce investigation made on privacy risk regarding the telecommunications sector, it is recommended that, in the future, its effect is analysed in different countries.

**Author Contributions:** Conceptualization, J.T. and S.T.; methodology, J.T. and S.T. validation, J.T. and S.T.; formal analysis, J.T. and S.T.; investigation, J.T. and S.T.; resources, J.T.; data curation, J.T.; writing—original draft preparation, J.T.; writing—review and editing, J.T. and S.T.; supervision, J.T. and S.T.; project administration, S.T.; funding acquisition, S.T. All authors have read and agreed to the published version of the manuscript.

**Funding:** This research was funded by FCT-FUNDAÇÃO PARA A CIÊNCIA E A TECNOLOGIA, grant number UIDB/05422/2020.

**Institutional Review Board Statement:** Not applicable. In Portugal, any study in which sensitive topics are not addressed and which excludes tests performed on humans (for example, drugs) does not require prior approval from the ethics board. Even so, ethical procedures generally accepted in social research were applied. The empirical study was anonymous, confidential and participation was voluntary. Each respondent gave informed consent for data collection and processing and future publication of results. Participants received information about (1) general study objectives, estimated time, and general participation characteristics; (2) the right to refuse to participate in the study and to discontinue participation at any time. No personal information was requested, and the data considered to characterize the sample do not allow for the identification of any participant. Thus, we believe that the rights of the respondents were assured.

**Informed Consent Statement:** Informed consent was obtained from all subjects involved in the study.

**Data Availability Statement:** Not applicable.

**Conflicts of Interest:** The authors declare no conflict of interest. The funders had no role in the design of the study; in the collection, analyses, or interpretation of data; in the writing of the manuscript; or in the decision to publish the results.

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
