# Peer review of "The Antecedents of Customer Satisfaction in the Portuguese Telecommunications Sector"

_sustainability, doi:10.3390/su15032778_

Round 1

Reviewer 1 Report

1.      It was with great pleasure that I read the article, which not only turned out to be very interesting but also very carefully written.

2.      The advantage of the presented article is a very clearly formulated research objective, a rich review of the literature (over 100 items in section references) on the subject and clearly formulated research hypotheses that were verified during the empirical study performed through Structural Equation Modelling (SEM).

3.      Hypothesis 3 and Hypothesis 4 sound the same in the content of the article (Figure 1 shows that they should sound differently).

4.      Also noteworthy is the very careful design of the main research tool with the use of Exploratory Factor Analysis (EFA), followed by a Confirmatory Factor Analysis (CFA) and reliability analysis. Measurement items in the questionnaire were selected on the basis of literature. My suggestion is that maybe it's worth considering new items that will be the result of author's thoughts?

5.      The authors also took care of a very clear presentation of the results (included tables facilitate the perception of results). Each of the results (verified hypothesis) is presented in the context of the findings made in the literature and the specificity of the Portuguese telecommunication market.

6.      The summary of the article contains suggestions for the Portuguese market of telecommunications services, which result from the results of the study. But what is also very important, the authors are aware of the limitations of the study. These limitations were indicated by the authors as directions for further research.

Reviewer 2 Report

The themes of customer satisfaction, loyalty, and trust are relevant and in the order of the day, because they are the guarantor of customer loyalty for companies and allow them the maintenance or growth of sales, so I consider it a current and interesting theme for both the academic and business community.

The paper is well structured. The abstract, introduction and literature review, hypothesis formulation, and discussion of the results are written in a thoughtful and balanced way. However, I suggest some changes, of the bibliography referenced throughout the paper only approximately 12% is from the last 5 years. I believe that this ratio should be no less than 30%, and the desirable ratio is equal to or greater than 50%, so I recommend that more current works should be referenced, increasing the ratio to values as close as possible to 30%.

This paper studies the determinants of loyalty and satisfaction of Portuguese customers in the telecommunications sector, developing a model that included customer perceptions of privacy risk and trust. The authors develop a model that includes customer perceptions of privacy and trust risk, a subject that has not been that studied. With this work, they also contribute to wider dissemination and development of knowledge on the subject, despite being only for the Portuguese market, a small market, which makes it difficult to deduce conclusions for other larger markets.

The methodology is presented in clearly and objective way, being correct to achieve the objectives of the work, the authors use structural equations to analyze the relationships between the different variables. 

The conclusion is presented in an objective and parsimonious way, without being long and tedious.

Reviewer 3 Report

The analysis is mostly clear and concise, but some changes and clarifications are still required: 

1) Provide more information about the sampling (sampling methods, response rate, timing) and the questionnaire itself. Without the questionnaire, it is not clear how the concepts used in the hypotheses have been operationalised. 

2) Reconsider Hypothesis 2 (which is not confirmed anyway) from the perspective of causality - as it seems rather counter-intuitive. Does trust really cause satisfaction or is it vice versa? People tend to trust because they are satisfied - do you really think they are satisfied because they trust? The previous research referenced by the authors confirms the relationship between these concepts but mostly not the direction of causality. 

3) Clarify Hypothesis 7 regarding the positive effects of privacy risk to trust. What does that mean precisely? Does it mean that if privacy risks at a certain company is higher, people will trust it more? I guess it is not meant that way. Please clarify. 
